# Automated Quality Control for Intravascular Optical Coherence Tomography with Limited Labels

**Yi Chen**[1]                                                    YI.CHEN@ADELAIDE.EDU.AU
**Jiawen Li**[1]                                               JIAWEN.LI01@ADELAIDE.EDU.AU
[1] *School of Electrical and Mechanical Engineering, Adelaide University, Australia*
**Minh-Son To**[2]                                         MINHSON.TO@FLINDERS.EDU.AU
[2] *Flinders Health and Medical Research Institute, Flinders University, Australia*
**Peter J. Psaltis**[3]                                    PETER.PSALTIS@ADELAIDE.EDU.AU
[3] *Lifelong Health Theme, South Australian Health and Medical Research Institute, Australia*

## Abstract

Artifacts in intravascular optical coherence tomography (IVOCT) imaging can obscure vessel-wall visualization and compromise downstream plaque assessment, yet automated quality control (QC) remains insufficiently developed. We present a label-efficient framework that retains expert-reviewed samples as anchors during progressive refinement of unlabeled data and uses a small calibration subset to derive class-specific decision thresholds, enabling more reliable decision-making under limited labels. On a clinical IVOCT dataset, it achieves strong artifact recognition together with a macro-specificity of 0.9536 and an expected calibration error (ECE) of 0.0678, with stronger specificity and calibration than seed-only supervision and generic semi-supervised learning baselines on the same backbone. These findings support automated QC as a practical front-end step for IVOCT analysis.

**Keywords:** intravascular optical coherence tomography, quality control, artifacts, label-efficient learning.

## 1. Introduction

Intravascular optical coherence tomography (IVOCT) provides high-resolution imaging for assessing coronary plaque morphology, but clinical pullbacks often contain frames degraded by artifacts such as blood, blooming, and sew-up (Wang et al., 2015; Iarossi Zago et al., 2020; Sperti et al., 2025). Although frame quality is routinely assessed by expert review in clinical IVOCT interpretation, automated quality control (QC) remains insufficiently developed for reliable and scalable downstream analysis (Wang et al., 2019; Jin et al., 2025). Developing robust QC models is challenging because expert annotation is costly (Jin et al., 2023) and artifact categories can affect image interpretability in different ways (Lindenholz et al., 2018).

To alleviate the reliance on dense expert review, label-efficient learning offers a practical direction for IVOCT QC (Li et al., 2024). However, many pseudo-labeling methods iteratively expand supervision using model predictions, which can accumulate errors and drift over refinement (Sohn et al., 2020; Wang et al., 2022; Karaliolios et al., 2023). This limitation is particularly relevant to IVOCT QC, where reliable screening may require more than uniform confidence-based selection. We therefore propose a framework that retains expert-reviewed samples as anchors during progressive refinement of unlabeled data and uses a held-out calibration subset to derive class-specific decision thresholds. On a clinical IVOCT dataset, the proposed framework achieves competitive artifact recognition together with improved specificity and calibration, supporting more reliable and label-efficient IVOCT quality screening.

## 2. Method

As illustrated in Fig. 1, we propose an anchor-stabilized progressive framework for label-efficient IVOCT quality control. The data are partitioned into an expert-reviewed seed set $\mathcal{D}_s$, a held-out calibration subset $\mathcal{D}_c$, and an unlabeled pool $\mathcal{U}$. In contrast to standard pseudo-labeling pipelines, the framework retains $\mathcal{D}_s$ as a persistent anchor set throughout refinement so that expert supervision remains active at every stage (Lee et al., 2013; Sohn et al., 2020). The held-out subset $\mathcal{D}_c$ is used for class-specific calibration and threshold selection. For each category $c$, a confidence threshold $\tau_c$ is selected to satisfy predefined reliability targets, such as minimum positive predictive value (PPV) or recall constraints (Fu et al., 2019).

Refinement proceeds through a Seed $\rightarrow$ Teacher $\rightarrow$ Student pipeline. A seed model is first trained on $\mathcal{D}_s$, calibrated on $\mathcal{D}_c$, and used to identify an initial reliable subset $\mathcal{R}^{(1)} \subset \mathcal{U}$. A teacher model is then trained on $\mathcal{D}_s \cup \mathcal{R}^{(1)}$, recalibrated on $\mathcal{D}_c$, and used to produce a refined reliable subset $\mathcal{R}^{(2)} \subset \mathcal{U}$. Finally, a student model is optimized on the reliable branch $\mathcal{D}_{\text{rel}}^{(2)} = \mathcal{D}_s \cup \mathcal{R}^{(2)}$ together with the remaining unlabeled pool $\mathcal{U}_{\text{rem}} = \mathcal{U} \setminus \mathcal{R}^{(2)}$:

$$\mathcal{L}_{stu} = \mathcal{L}_{rel}(\mathcal{D}_{\text{rel}}^{(2)}) + \lambda_u \mathcal{L}_{unsup}(\mathcal{U}_{\text{rem}}).$$

Here, $\mathcal{L}_{rel}$ uses expert labels for $\mathcal{D}_s$ and teacher-generated targets for $\mathcal{R}^{(2)}$, while $\mathcal{L}_{unsup}$ regularizes the remaining unlabeled pool. This progressive design enables label-efficient refinement with persistent expert anchors and calibration-guided pseudo-label selection for IVOCT quality screening.

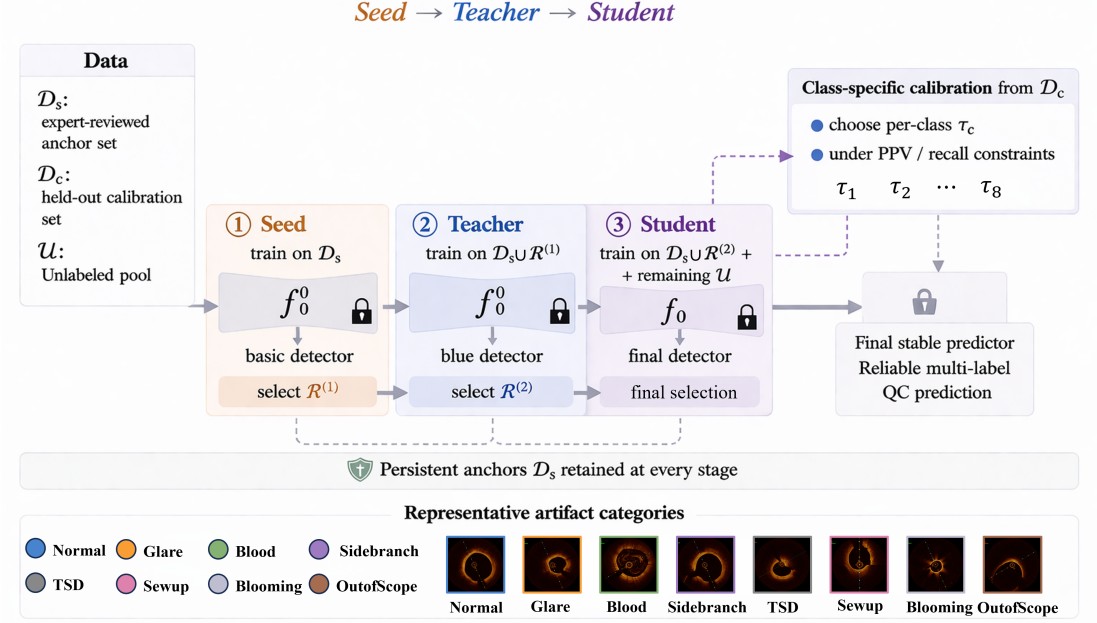

Figure 1: Overview of the proposed label-efficient quality-control framework for IVOCT, with persistent anchors and class-specific calibration. TSD: Tangential signal dropout (Araki et al., 2022).

## 3. Experiments and Results

We evaluate the proposed framework on a clinical IVOCT dataset from the Coronary Plaque Modification in Acute Coronary Syndrome (COCOMO-ACS) study (Montarello et al., 2022; Psaltis et al., 2025). From this cohort, we extracted 130 pullbacks from 64 patients. Data are split at the pullback level, with all frames from each pullback assigned to a single subset. The dataset includes an expert-reviewed seed set $\mathcal{D}_s$ of 898 labeled frames, accounting for about 3.7% of the training pool, a held-out calibration set $\mathcal{D}_c$, an unlabeled pool $\mathcal{U}$, and separate validation and test sets. QC is formulated as an 8-class multi-label artifact detection task. All models use backbones from the ConvNeXtV2 family (Woo et al., 2023). Performance is evaluated using macro-F1, macro-specificity, macro-averaged area under the precision–recall curve (AUPRC-Ma), and macro-averaged area under the ROC curve (AUC-Ma), together with Expected Calibration Error (ECE). Table 1 summarizes the main results. On the ConvNeXtV2-Tiny backbone, the proposed framework achieves the best macro-specificity (0.9536) and ECE (0.0678) among the compared methods, while maintaining competitive macro-F1 (0.6577) and AUPRC-Ma (0.7421).

Table 1: Quantitative comparison of the proposed framework.

| Group | Method | Backbone | Ma-F1 | Ma-Spec | AUPRC-Ma | AUC-Ma | ECE |
|---|---|---|---|---|---|---|---|
| Baseline | Supervised ($\mathcal{D}_s$) | ConvNeXtV2-T | 0.6544 | 0.9157 | 0.7499 | 0.9196 | 0.0752 |
| | Supervised ($\mathcal{D}_s$) | ConvNeXtV2-B | 0.6323 | 0.9283 | **0.7652** | **0.9309** | 0.0807 |
| Generic SSL | FixMatch | ConvNeXtV2-B | **0.6654** | 0.9373 | 0.6853 | 0.8974 | 0.0810 |
| | FreeMatch | ConvNeXtV2-B | 0.6450 | 0.9376 | 0.6805 | 0.8950 | 0.0865 |
| | FixMatch | ConvNeXtV2-T | 0.6431 | 0.9366 | 0.6829 | 0.8922 | 0.0761 |
| | FreeMatch | ConvNeXtV2-T | 0.6618 | 0.9326 | 0.6987 | 0.9064 | 0.0794 |
| **Proposed** | **Reli-OCT (Ours)** | ConvNeXtV2-T | 0.6577 | **0.9536** | 0.7421 | 0.9254 | **0.0678** |

## 4. Discussion and Conclusion

The results suggest that a small amount of expert-reviewed data already provides a strong starting point for IVOCT QC. In particular, the ConvNeXtV2-Base supervised baseline achieved the strongest AUPRC-Ma and AUC-Ma, indicating that backbone capacity remains important for discrimination performance under limited IVOCT labels (Chen et al., 2020; Wang et al., 2023). By contrast, the proposed framework achieved the best macro-specificity and ECE on the ConvNeXtV2-Tiny backbone, suggesting that its main benefit lies in improving reliability-oriented screening behavior. This pattern is consistent with the role of QC as a screening step, where reducing false positives and improving calibration are both important. Future work will investigate how to preserve reliability while enabling smaller backbones to approach or surpass stronger backbones in broader clinical settings.

### Acknowledgments

We gratefully acknowledge Giuseppe Di Giovanni for his guidance in training and supporting the annotation process. We also sincerely thank Harry Carpenter for his assistance in facilitating the ethical approval process.

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
