# OpenReview forum: "Automated Quality Control for Intravascular Optical Coherence Tomography with Limited Labels"
_MIDL.io/2026/Short_Papers — MIDL 2026 - Short Papers Poster_

### Official Review · Reviewer_PJHo · 2026-05-03
**Interesting QC method, with minor clarity issues. Some claims are not fully supported by experiments.**

**Rating:** 4
**Confidence:** 3

**Review:**

See strengths and weaknesses.

**Summary:**

The paper targets automated quality control of IVOCT pullbacks, where artifacts (blood, blooming, sew-up, tangential signal dropout, etc.) can corrupt downstream plaque analysis. Expert annotation is the bottleneck. The proposed approach incorporates several components that (i) keeps the small expert-labeled seed set as a persistent anchor through every refinement stage, (ii) uses a held-out calibration subset Dc to set per-class confidence thresholds τ_c that meet predefined PPV/recall targets, and (iii) trains the final student with a weighted combination of labeled+pseudo-labeled "reliable" samples and an unsupervised regularizer on the remainder. Evaluation includes 130 pullbacks / 64 patients from COCOMO-ACS. Reli-OCT on ConvNeXtV2-T reaches the best macro-specificity (0.9536) and ECE (0.0678) among compared methods, with macro-F1 0.6577 and AUPRC-Ma 0.7421, beating FixMatch and FreeMatch on calibration but not uniformly on discrimination.

**Strengths:**

1. The problem is clinically relevant and well motivated. IVOCT QC really is a bottleneck, and per-frame artifact gating matters for downstream plaque models. The framing as a "front-end screening step where false positives and calibration matter most" is appropriate and the metrics chosen (specificity + ECE) align with that framing.

2. I like the persistent-anchor idea. Real failure mode of self-training (drift/confirmation bias) is a common problem, and the separation of pseudo-label selection from threshold calibration via a held-out Dc is a nice methodological choice.

**Weaknesses:**

1. I found the headline result is somehow narrower than the abstract suggests. The abstract says the method has "stronger specificity and calibration than seed-only supervision and generic semi-supervised learning baselines on the same backbone." However, I found in Table 1:
    * Macro-F1: Reli-OCT (T) 0.6577 is *worse* than FixMatch-B 0.6654 and FreeMatch-T 0.6618.
    * AUPRC-Ma: Reli-OCT (T) 0.7421 is *worse* than both supervised baselines (T 0.7499, B 0.7652).
    * AUC-Ma: Reli-OCT (T) 0.9254 is *worse* than supervised-T 0.9196? Actually slightly better (.9254 > .9196), but worse than supervised-B 0.9309.
In other words, Reli-OCT is NOT always the best. It wins on specificityand ECE but trades discrimination. This is a defensible position for a screening QC tool, but the abstract phrasing oversells it. Typically for a MIDL short paper people are not expecting comprehensive/thorough experiments due to page limit, but I encourage the authors to revise the narrative to be self coherent.

2. There are some clarity issues. There are some key methodological details are absent or hand-waved:
    * The unsupervised loss L_unsup is mentioned but never defined.
      Is it consistency regularization FixMatch? Entropy
      minimization? It matters.
    * λ_u is mentioned but its value is not given.
    * "Predefined reliability targets" for τ_c (minimum PPV / recall)
      are referenced but the actual targets used are not specified.
    * It is unclear whether τ_c is set to target the same operating
      point as the baselines' default. if not, the specificity/ECE
      comparison is partly a comparison of operating points, not of
      methods.
  * The 8-class multi-label setup needs more reporting. Per-class
  performance, especially for rare artifact classes (e.g., bubbles or
  TSD), would matter clinically far more than macro-averages. The
  paper reports only macro statistics, which can hide catastrophic
  drops on minority classes. It is also not clear to me what exactly are the eight classes (artifacts); the authors only reported four.

3. The paper titles QC but is somewhat doing artifact flagging task. They call it "QC" (which implies a use/don't-use decision) but train it as artifact detection (which is a multi-label tagging task). The mapping from "frame X has blooming + blood" to "frame X is unusable" is never specified. There is no binary acceptable/unacceptable head.

**Justification Of Rating:**

The paper solves a clinically relevant problem with some thoughtful design choices. The evaluation is relatively coherent and thorough, clearly above the bar of a short paper track. That said, there are some limitations that should be addressed should the paper be accepted by PC. For example, the paper names only four artifacts explicitly: blood, blooming, sew-up, and tangential signal dropout (TSD). The other four are never listed in the main text. and there are some over claimed arguments in the abstract.

Nevertheless, I believe these weaknesses are mainly clarification/editorial fixes that can be addressed in the camera ready version. Overall the merits outweigh these weaknesses.

---

### Decision · Program_Chairs · 2026-05-08

Accept (Poster)